# Over- and Undercoordinated Atoms as a Source of Electron and Hole Traps in Amorphous Silicon Nitride (a-Si_3_N_4_)

**DOI:** 10.3390/nano13162286

**Published:** 2023-08-09

**Authors:** Christoph Wilhelmer, Dominic Waldhoer, Lukas Cvitkovich, Diego Milardovich, Michael Waltl, Tibor Grasser

**Affiliations:** 1Christian Doppler Laboratory for Single-Defect Spectroscopy in Semiconductor Devices, Institute for Microelectronics, TU Wien, 1040 Wien, Austria; 2Institute for Microelectronics, TU Wien, Gusshausstrasse 27-29, 1040 Wien, Austriagrasser@iue.tuwien.ac.at (T.G.)

**Keywords:** amorphous silicon nitride, intrinsic charge trapping sites, flash memory, nonradiative multi-phonon model

## Abstract

Silicon nitride films are widely used as the charge storage layer of charge trap flash (CTF) devices due to their high charge trap densities. The nature of the charge trapping sites in these materials responsible for the memory effect in CTF devices is still unclear. Most prominently, the Si dangling bond or *K*-center has been identified as an amphoteric trap center. Nevertheless, experiments have shown that these dangling bonds only make up a small portion of the total density of electrical active defects, motivating the search for other charge trapping sites. Here, we use a machine-learned force field to create model structures of amorphous Si${}_{3}$N${}_{4}$ by simulating a melt-and-quench procedure with a molecular dynamics algorithm. Subsequently, we employ density functional theory in conjunction with a hybrid functional to investigate the structural properties and electronic states of our model structures. We show that electrons and holes can localize near over- and under-coordinated atoms, thereby introducing defect states in the band gap after structural relaxation. We analyze these trapping sites within a nonradiative multi-phonon model by calculating relaxation energies and thermodynamic charge transition levels. The resulting defect parameters are used to model the potential energy curves of the defect systems in different charge states and to extract the classical energy barrier for charge transfer. The high energy barriers for charge emission compared to the vanishing barriers for charge capture at the defect sites show that intrinsic electron traps can contribute to the memory effect in charge trap flash devices.

## 1. Introduction

Flash memory devices are usually assembled as stacked layers of different materials, for example, as silicon–oxide–nitride–oxide–silicon (SONOS) or tantalum nitride–aluminum oxide–nitride–oxide–silicon (TANOS) structures, where silicon nitride is implemented as the charge storage layer. Silicon nitrides are the material of choice for these applications due to their high trap densities (≈$10^{19}$ cm${}^{-3}$ [1,2]). By applying a large positive gate voltage to the device, electrons can cross the tunnel oxide via Fowler–Nordheim tunneling and localize at a specific trapping site in the nitride, thereby changing the electrostatics of the device and allowing for the realization of two or more different states (programming cycle). Vice versa, the states can be erased by applying a large negative voltage, pushing back the stored electrons to the substrate (erase cycle). Furthermore, holes from the substrate can be attracted and localize at specific trapping sites in the nitride layer [3,4].

The nature of the sites responsible for electron and hole trapping is still under discussion. In the past, the dangling Si bond or *K*-center has been the target of several experimental and theoretical studies [5,6,7,8], being suspected as the main contributor to the charge trapping mechanism in amorphous silicon nitrides. However, experimental investigations of the defect concentration (≈$10^{16}$–$10^{18}$ cm${}^{-3}$ [1,9,10]) show that these dangling bonds only make up a small portion of the total density of electrical active defects. Furthermore, theoretical investigations of electronic states introduced by over- and under-coordinated atoms in Si${}_{x}$N${}_{y}$:H [11,12] and the localization of electrons in amorphous Si${}_{3}$N${}_{4}$ (a-Si${}_{3}$N${}_{4}$) at these sites [7] indicate a variety of possible charge trapping centers in amorphous silicon nitrides.

Here, we investigate the localization of holes and electrons at intrinsic trapping sites in pure a-Si${}_{3}$N${}_{4}$ using ab initio methods. The a-Si${}_{3}$N${}_{4}$ structures studied in this work are created with molecular dynamics (MD), which has successfully been used in the past to model amorphous bulk materials and study structural and electronic properties of nanostructures [13,14,15]. We employ a machine-learned force field trained on data from density functional theory (DFT) calculations to accurately sample the potential energy surface of the material. The amorphous structures are created by simulating a melt-and-quench procedure with MD and are subsequently further relaxed with DFT employing a hybrid functional. We show that in a-Si${}_{3}$N${}_{4}$, additional charges can localize near over- or under-coordinated atoms, which are present in this material in considerable portions. Namely, holes preferably localize near two-fold coordinated N or five-fold coordinated Si, while electrons localize near three-fold coordinated Si or four-fold coordinated N. The trapping sites are statistically investigated by analyzing defect states in the band gap after capturing a charge and relaxation of the structure. The charge trapping sites are characterized in the context of a nonradiative multi-phonon (NMP) model [16,17] by calculating their relaxation energies $E_{\mathrm{Relax}}$ and thermodynamic charge transition levels (CTLs) to model the potential energy curve (PEC) of the systems in different charge states. Subsequently, the classical charge transition barriers are extracted from the crossing points of the PECs for phonon-driven charge capture and emission processes to show that intrinsic trapping sites in a-Si${}_{3}$N${}_{4}$ are suitable candidates to trap and store electrons from Si substrates and thus can contribute to the memory effect in charge trap flash devices.

## 2. Materials and Methods

In this section, the computational setup for the creation of our model structures and for the analysis of the electronic states is described. The structural properties of our a-Si${}_{3}$N${}_{4}$ samples are analyzed and the NMP model used to characterize the charge transfer processes at intrinsic trapping sites is described.

### 2.1. Computational Setup

Our amorphous model structures were created by simulating a melt-and-quench procedure within classical molecular dynamics (MD), employing a machine-learned interatomic potential (ML-IP). This potential was specifically trained to create amorphous structures from a melt-and-quench procedure. To account for the structural variety of amorphous systems, the ML-IP training set consisted of energies, forces and stress tensors calculated with DFT for more than 1600 different Si${}_{3}$N${}_{4}$ structures. The structures of the training data set included amorphous systems with different mass densities, dimers and different crystalline phases of Si${}_{3}$N${}_{4}$. The training procedure and validity of this force field which uses an efficient active learning technique, combined with the Gaussian approximation potential (GAP) [18] and three different descriptors, is discussed in detail in [19]. The GAP method computes the total potential energy of a given atomic system as the sum of a local energy contribution from each atom and has been successfully used for the development of different ML-IPs in the recent past [20,21]. All MD calculations were performed using the LAMMPS code [22] in conjunction with the QUIP package, using a timestep of 0.5 fs for the Verlet integration and employing a Langevin thermostat to control the temperature of the system. For the DFT calculations, we employed the Gaussian plane wave (GPW) method as utilized in the CP2K code [23] in conjunction with the Goedecker–Teter–Hutter (GTH) pseudopotentials. To accurately describe the electronic wave function, we use a double-zeta Gaussian basis set in conjunction with the non-local hybrid functional PBE0_TC_LRC [24] and the auxiliary basis set pFIT3, which was successfully employed in recent studies to investigate intrinsic charge trapping sites [25] to reduce the computational costs of calculating the Hartree–Fock exchange. The total energy of geometry optimizations and single point calculations was converged self-consistently down to 2.7 µeV.

### 2.2. Structure Creation and Characterization

Due to the structural randomness of a-Si${}_{3}$N${}_{4}$, investigations of the charge trapping properties have to be performed in a statistical manner, as each trapping site is expected to show different defect properties depending on the local environment. Therefore, 100 initial a-Si${}_{3}$N${}_{4}$ structures with 224 atoms each were created by simulating a melt-and-quench procedure, as has already been successfully used to model other amorphous compounds [14,26,27]. The initial Si${}_{3}$N${}_{4}$ samples were heated above their melting points to 5000 K and kept at this temperature for 60 ps until they lost all initial information. The samples were then slowly cooled down to room temperature (300 K) with different quenching rates ranging from 0.1 to 5 K ps${}^{-1}$, resulting in several unique amorphous Si${}_{3}$N${}_{4}$ structures. Subsequently, the structures were geometry optimized with DFT in charge states *q* = 0, −1 and +1 to calculate the minimum energy configuration in each charge state.

Our model structures correctly reproduce the key structural properties of silicon nitride thin films. The structure factor *S* and radial distribution function g(r) of a model structure were compared to sample characterizations from scattering experiments [28,29] in Figure 1a,b. The average experimental Si-N bond length of 1.75 Å (indicated by the green vertical line in Figure 1c) coincides with the peak of the Si-N bond length distribution over all generated structures. The tail of the distribution at higher values can be attributed to slightly strained bonds in the amorphous network and is in agreement with previous theoretical studies [13,30]. N-N bonds and Si-Si bonds were completely absent in all of our quenched structures due to the high energies associated with these structures in the training data set of the ML-IP. The mass density of the structures is 2.93 g cm^−3^, which is in the range of experimental values varying from 2.3 to 3.0 g cm^−3^ [31].

In ideal stoichiometric Si${}_{3}$N${}_{4}$, every Si in the network has four neighboring N atoms, while every N has three neighboring Si atoms. In our amorphous model structures, 1.5–3.5% N and Si are over- or under-coordinated on average, similarly as reported in structures created by a classical force field [11]. The energy gap between the highest occupied molecular orbital (HOMO) and lowest unoccupied molecular orbital (LUMO) depends on the relative number of undercoordinated atoms, as shown in Figure 1d. With an increasing number of undercoordinated atoms in a single structure, the dangling bonds are increasingly likely to introduce states in the band gap, resulting in a reduced HOMO–LUMO gap. In the following, only structures with a well-defined band gap above 3.8 eV, as denoted with a red horizontal line in Figure 1d, without any states in between are investigated to ensure sufficient quality of the structures and to allow for the analysis of specific single charge trapping processes. The band gaps of the further investigated 24 structures are distributed around $E_{\mathrm{GAP}}=4.08\pm 0.25$ eV, which agrees well with previous DFT calculations [11], but underestimates the experimentally determined band gap values from the literature, ranging from 4.5 to 5.3 eV [1]. This underestimation of the band gap is typical for DFT calculations employing hybrid functionals without explicitly tuning the mixing parameter α to the experimental bandgap [32].

### 2.3. Charge Trapping Model

Numerous studies have shown that charge trapping processes can be accurately described by a nonradiative multi-phonon model [16,17] with the most important parameters of the model, namely the relaxation energies $E_{\mathrm{Relax}}$ and trap level $E_{T}$, schematically depicted in Figure 2.

In the classical limit of NMP theory, which is usually applicable at room temperature or above [33], the charge transfer rate is governed by the crossing point of the PECs, defining a classical barrier as denoted as the transition level NMP in Figure 2. The PECs in both charge states are approximated by a parabolic function near their minimum energy configurations which is commonly used and known as the harmonic approximation (HA) [17,34]. Within this approximation, only three points of the potential energy surface have to be calculated in order to obtain the forward and backward energy barriers. Compared to the minimum energy path (MEP), extractions of energy barriers within the HA are computationally feasible even for statistical analysis in complex amorphous systems. Hereby, we have not performed a second order expansion of the potential energy surface in the minimum, e.g., calculating an effective phonon frequency based on the curvature in the minimum, but rather have used points which are defined by the relaxation energies. The HA description offers a more global approximation of the PES and is usually in good agreement with the MEP [35], governing the charge trapping dynamics at room temperature and above. Due to thermal excitation, the system can be forced out of its minimum energy configuration and capture or emit a charge from, e.g., a Si substrate by overcoming the classical energy barrier to the crossing point and subsequently relaxing to a new minimum energy configuration in the new charge state. The PECs are given as a function of the configuration coordinate, representing the structural change of the total system projected on an effective 1D reaction coordinate, and are drawn in the context of a Si/Si${}_{3}$N${}_{4}$ band diagram, where the conduction band minimum (CBM) acts as the charge reservoir for electron traps (Figure 2 (left)). Vice versa for the hole trapping mechanism, the valence band maximum (VBM) serves as the hole reservoir as shown in Figure 2 (right). Hence, only interactions of the defects with the band edges are analyzed in this study. As discussed in [36], in the strong electron–phonon coupling regime, the energy barriers extracted by employing this approximation agree very well with the energy barrier obtained when the full density of electronic states is considered. For both cases, the minimum energy of one of the parabolas is pinned to the respective charge reservoir, corresponding to the total energy of the system when the transferred electron is still in the substrate. When a gate bias is applied, the bands of Si${}_{3}$N${}_{4}$ are shifted due to the field in the nitride. This also results in a shift in the parabolas against each other by δ and thus the stability of each charge state and the respective barriers to the NMP transition level are changed.

The stability of the respective charge state can be analyzed by calculating the formation energy of a defect system. For a system in charge state *q*, the formation energy $E_{\mathrm{Form}}^{q}$ is given by
(1)$$E_{\mathrm{Form}}^{q}=E_{\mathrm{tot}}^{q}-E_{\mathrm{tot}}^{\mathrm{bulk}}-\sum_{i}\mu_{i}n_{i}+qE_{F}+E^{\mathrm{corr}}$$
where $E_{\mathrm{tot}}^{q}$ is the total energy of the defect system, $E_{\mathrm{tot}}^{\mathrm{bulk}}$ is the total energy of a reference system such as a pristine bulk and $\sum_{i}\mu_{i}n_{i}$ corresponds to the energy needed to add or remove *n* atoms of type *i* with the chemical potential μ. $E^{\mathrm{corr}}$ is a correction term needed for DFT calculations for charged systems with periodic boundary conditions to remove any artificial interactions of the charged defect with its periodic image. This term was calculated as $E^{\mathrm{corr}}\approx 0.12$ eV for our systems employing the sxdefectalign tool, which implements the Freysoldt–Neugebauer–Van de Walle correction scheme [37]. $E_{F}$ corresponds to the energy of the transferred charge carrier, given with respect to the highest occupied Kohn–Sham state $E_{F}=E_{\mathrm{VBM}}+\epsilon_{F}$. When comparing the formation energies of the charged and uncharged systems, the energy of the reference system $E_{\mathrm{tot}}^{\mathrm{bulk}}$ cancels out and $\sum_{i}\mu_{i}n_{i}=0$ as no structural defect is created. The CTL corresponds to the energy of the transferred charge carrier where the formation energies of the charged and uncharged structure are equal $E_{\mathrm{Form}}^{q_{1}}\left(E_{F}\right)=E_{\mathrm{Form}}^{q_{2}}\left(E_{F}\right)$. It can be related to the trap level by $E_{T}=\mathrm{CTL}-E_{F}$. To date, only the energetic positions of the minimum configurations were considered. To study the charge trapping dynamics however, additional information about the PECs is required. This is provided by the relaxation energies $E_{\mathrm{Relax}}$, which are a measure for the curvature of the PECs near the minimum. Furthermore, for optical transitions, $E_{\mathrm{Relax}}$ corresponds to the energy difference between a given initial defect state and its relaxed configuration after an immediate (un)charging event. With these parameters, the PECs in two charge states can be modeled relative to each other and the energy barriers for charge capture and emission can be extracted from the crossing point.

## 3. Results and Discussion

In the following, the intrinsic charge trapping sites in a-Si${}_{3}$N${}_{4}$ are characterized according to their structural features, electronic states and thermodynamic properties. The structures are geometry optimized in different charge states and the electronic wave functions and eigenenergies of the Kohn–Sham states are calculated with DFT. First, the density of states of the model structures before and after the charge trapping process including structural relaxation are analyzed. Subsequently, the relaxation energies, charge transition levels and charge transition barriers of the charge trapping sites are calculated in the context of the NMP model.

### 3.1. Trapping Sites

In this section, the localization of charge near over- and under-coordinated atoms and the projected density of states (PDOS) before and after a single charge capture event in a-Si${}_{3}$N${}_{4}$ are analyzed.

#### 3.1.1. Hole Traps

The PDOS of two different a-Si${}_{3}$N${}_{4}$ structures before and after trapping a hole is shown in Figure 3a,b (bottom), with the trapping sites depicted above the PDOS before and after a single hole is captured.

Positive and negative values of the PDOS correspond to the majority and minority spin channels, respectively. We find that electronic states near the valence band maximum (VBM) are introduced either by overcoordinated Si or undercoordinated N, which is in agreement with previous theoretical and experimental studies [11,12,38]. In the first case, the HOMO is semi-localized and hybridized around the five adjacent N atoms of the overcoordinated Si, as shown in Figure 3a (top), with the isosurface of the orbital drawn at a value of 0.05 e/Å${}^{3}$. When a hole is introduced to the system, the structure relaxes to a new minimum energy configuration, thereby shifting the state at the VBM, now unoccupied, towards the middle of the band gap. Similarly, in the second structure, where the HOMO localizes at an undercoordinated N, a hole can be trapped at the two-fold coordinated N, thereby shifting the now unoccupied state towards the middle of the band gap as shown in Figure 3b.

#### 3.1.2. Electron Traps

The PDOS of two different a-Si${}_{3}$N${}_{4}$ structures before and after capturing an electron is shown in Figure 4a,b (bottom), with the trapping sites plotted above the PDOS before and after a single electron is trapped. Depending on the respective structure, the LUMO orbital is either localized at a Si near a four-fold coordinated N or hybridized between an undercoordinated and a fully coordinated Si, which also agrees with previous findings [11,12]. After trapping an electron near a four-fold coordinated N, the Si-N distance increases, thereby shifting the state at the conduction band minimum (CBM) towards the middle of the band gap. For some cases, the LUMO is hybridized between a three-fold and a fully coordinated Si, as shown in Figure 4b. After trapping an electron, these Si atoms move closer together, thereby introducing two states in the band gap, one occupied and one unoccupied.

#### 3.1.3. Defect States

The energies of the Kohn–Sham states introduced in the band gap by adding a hole or an electron to the system are shown in Figure 5. The energies are given with respect to the VBM of each structure. A normal distribution was fitted to the energies with the fitting parameters shown in the plot. The CBM is given as a band with the energetic distance to the VBM according to the distribution of HOMO–LUMO gaps of the analyzed structures. The energy distribution of the occupied states after electron capture is slightly broader and lower in energy compared to the distribution of unoccupied states after trapping a hole.

#### 3.1.4. Inverse Participation Ratio

The inverse participation ratio (IPR) was calculated to analyze the degree of localization of Kohn–Sham states. The IPR, which has already successfully been used to characterize amorphous systems in the past [13,27], is a measure of the localization of a state and can be comfortably calculated by taking advantage of the atom-centered basis functions $\phi_{i}$ implemented in CP2K. A Kohn–Sham state $\psi_{n}$ can be described by a linear combination of the atom-centered basis functions $\psi_{n}=\sum_{i}^{N}c_{ni}\phi_{i}$, with N being the total number of atomic basis functions. The IPR of the Kohn–Sham state *n* is then given by
(2)$$\mathrm{IPR}\left(\psi_{n}\right)=\frac{\sum_{i}^{N}c_{ni}^{4}}{{\left(\sum_{i}^{N}c_{ni}^{2}\right)}^{2}}$$

For a fully delocalized state, the IPR has a comparably small value, while it increases for decreasing spatial extensions of the state. The IPR was calculated for two model structures: one where the states at the band edges correspond to overcoordinated atoms as shown in Figure 6a and one where the band edges correspond to undercoordinated atoms as shown in Figure 6b.

For both cases, the states at the band edges have far higher IPR values than states deeper in the valence or higher in the conduction band. States at undercoordinated atoms have noticeably higher IPR values compared to states at overcoordinated atoms, suggesting a stronger localization at the intrinsic defect site.

### 3.2. Defect Characterization

In the following, the charge trapping sites are analyzed according to the NMP model as described in Section 2.3.

#### 3.2.1. Charge Transition Level

The CTLs for hole and electron capture at the intrinsic trapping sites are shown in Figure 7 in the context of a Si/Si${}_{3}$N${}_{4}$ band diagram.

All CTLs are given with respect to the VBM of the respective a-Si${}_{3}$N${}_{4}$ model structure and are located inside the band gap of a-Si${}_{3}$N${}_{4}$. In contrast to the Kohn–Sham states introduced by a trapped charge as shown in Figure 5, the CTL is a thermodynamic property of a defect site (see Equation (Equation 1)). The valence band offset between Si${}_{3}$N${}_{4}$ and Si of 1.78 eV is taken from X-ray photoelectron spectroscopy experiments on Si(111)/Si${}_{3}$N${}_{4}$ samples as reported in [39]. Normal distributions were fitted to the CTLs with the fitting parameters given in the plot. CTLs for hole transfer are narrowly distributed 1.1 eV below the VBM of the Si substrate and can therefore only be charged by applying a negative voltage to the gate of a memory device. This agrees well with experimentally determined hole trap levels of 0.5 and 1.1 eV, measured for low-pressure chemical vapor deposition (LPCVD)-generated Si${}_{3}$N${}_{4}$ samples as reported in [40]. The distribution of the CTLs for electron transfer is broader, similarly to the introduced defect states in the band gap as shown in Figure 5 and located around the CBM of the Si substrate. The CTLs for trapping electrons compare well with theoretical literature values of intrinsic electron traps distributed around 1.5 eV below the CBM [7] and experimental values from trap spectroscopy by charge injection and sensing (TSCIS) measurements ranging between 0.8 and 1.8 eV below the CBM [41]. Hence, most of these sites can easily trap electrons from the substrate, while defect sites with CTLs above the CBM only efficiently capture charges with the Si substrate if the energy of the localized state is shifted towards the Si band edge by an external electric field. The field can be generated by an applied voltage on the gate of, e.g., a SONOS device. Thus, the absolute change in energy δ depends on the position of the state in the nitride.

#### 3.2.2. Relaxation Energy

Relaxation energies according to the notation in Figure 2 were calculated by performing single point calculations on relaxed configurations in different charge states and subsequently calculating the energy difference to the energy of the relaxed system in the same charge state. The relaxation energies for hole capture, hole emission, electron capture and electron emission are shown in Figure 8a–d with the parameters of fitted normal distributions given in the plots. The distributions of the relaxation energies show that for charge transfer processes involving electron transfer, the energies gained by structural relaxations of the systems are roughly 0.5 eV higher than for hole transfer. Furthermore, the energy distributions involving electron transfer are again broader compared to the energy distributions for hole transfer.

#### 3.2.3. Energy Barriers

Here, the defect parameters presented in the previous subsections are used to model the PECs shown in Figure 2 to extract the energy barriers ΔE from the minimum energy configurations to the transition level NMP at the crossing points of the PECs. The resulting energy barriers are shown as a correlation plot in logarithmic scale for charge capture and charge emission in Figure 9.

The energy barriers for electron transfer, which are shown in Figure 9a, are given with respect to an electron reservoir at the CBM of a Si substrate (δ=0 eV). By applying a positive voltage to the gate of, e.g., a SONOS device, the Si${}_{3}$N${}_{4}$ bands and thus the CTL with respect to the charge reservoir are shifted to lower values by δ. The energy barriers are also plotted for a shift of δ=−1 eV in Figure 9a. Compared to the initial conditions, for which the energy barriers are shown as turquoise circles, the barriers for electron emission significantly increase to higher values up to 2 eV, while the energy barriers for electron capture almost vanish. Thus, by applying a positive voltage, the intrinsic defect sites can easily trap and store electrons from the CBM of Si. The energy barriers for hole transfer are shown in Figure 9 (right) with the VBM acting as a hole reservoir. Initially, ΔE for hole capture is clearly higher than for hole emission, showing that without an applied bias the intrinsic hole trapping sites are rather unlikely to trap a hole from the Si VBM. The barriers are also plotted for negative bias conditions, changing the energetic difference between CTL and Si VBM by δ=+1 eV due to the electric field in the oxide. ΔE for hole capture is thereby reduced, while ΔE for hole emission is slightly increased. Compared to electron transfer, the barriers for emitting holes are still significantly lower, resulting in reduced storage time of holes at the intrinsic trapping sites. Due to the small energy barriers and CTLs close to the VBM, the investigated intrinsic hole trapping sites could be related to the experimentally determined hole traps in [40]. In this work, the authors suspect that phonons do not play a significant role in the hole trapping mechanism, which corresponds to low energy barriers in the classical NMP model.

## 4. Discussion and Conclusions

We statistically analyzed intrinsic electron and hole trapping sites in amorphous Si${}_{3}$N${}_{4}$ using a machine-learned force field for structure creation with molecular dynamics and ab initio methods to calculate the total electronic wave function. We show that states at the conduction band edge are introduced by overcoordinated N and undercoordinated Si, while states at the valence band edge correspond to overcoordinated Si and undercoordinated N. When a charge is added to the system, it localizes at one of these sites, thereby shifting the state towards the middle of the band gap due to structural relaxations. The trapping sites are characterized according to a nonradiative multi-phonon (NMP) model by calculating relaxation energies and charge transition levels to model the potential energy curves (PECs) of the sites in different charge states and to extract energy barriers for charge emission and capture from the crossing points of the PECs. We thereby show that intrinsic trapping sites can easily capture and store electrons from a Si substrate as, for example, realized in silicon–oxide–nitride–oxide–silicon (SONOS) stacks used in non-volatile flash memory devices. These parameters are also relevant for describing the charge trapping in device level models (e.g., technology computer-aided design (TCAD)). By applying a positive voltage to the gate of a SONOS device, which is typically done during the programming cycle, the energy barriers and thus the time constants for charge emission increase, while the barriers for charge capture decrease according to the NMP model. This shows that overcoordinated N and undercoordinated Si are suitable intrinsic defect candidates to contribute to the memory effect in non-volatile flash memory devices. We furthermore showed that although holes can be captured from the Si substrate when a negative gate voltage is applied, energy barriers for hole emission are significantly lower compared to the barriers for emitting electrons, resulting in a comparably reduced storage capability for holes in a-Si${}_{3}$N${}_{4}$.

## Figures and Tables

**Figure 1 nanomaterials-13-02286-f001:**
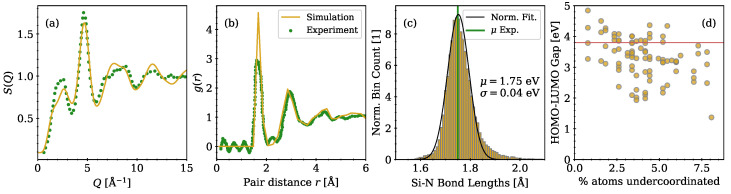
Structural properties of amorphous Si${}_{3}$N${}_{4}$ model structures compared with experimental data [28,29]. (**a**) Structure factor; (**b**) radial distribution function; (**c**) Si-N bond length distribution of all structures combined with a fitted normal distribution without stretched Si-N bonds > 1.85 Å. Fitting parameters are given in the plot and the green line denotes the mean bond length from experimental data. (**d**) Energy gaps between highest occupied molecular orbital (HOMO) and lowest unoccupied molecular orbital (LUMO) as a function of the relative number of undercoordinated atoms in the respective structure. Only structures with band gaps above 3.8 eV, as denoted with a red horizontal line, were further investigated.

**Figure 2 nanomaterials-13-02286-f002:**
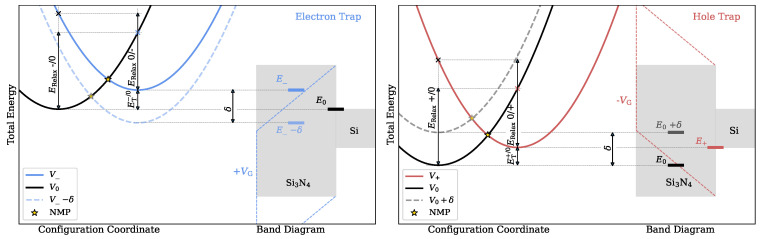
Schematics of potential energy curves as a function of the configuration coordinate for a Si${}_{3}$N${}_{4}$ system in two charge states in the context of a Si/Si${}_{3}$N${}_{4}$ band diagram to explain the charge transfer mechanism of electron traps (**left**) and hole traps (**right**). Applying a gate voltage shifts the trap level by δ as outlined with the dashed colored lines for positive (**left**) and negative (**right**) bias.

**Figure 3 nanomaterials-13-02286-f003:**
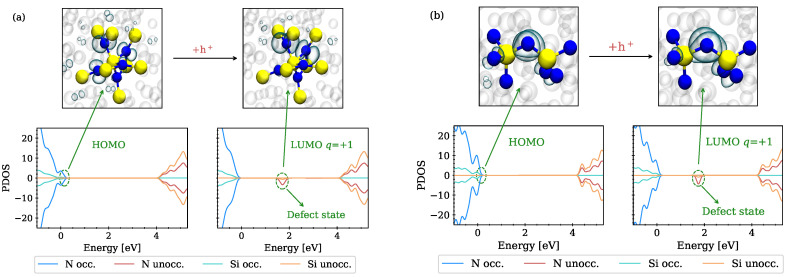
Hole trapping near intrinsic sites in a-Si${}_{3}$N${}_{4}$ with the localized wave functions (**top**) and the projected density of states of the structure (**bottom**). Silicon atoms are shown in yellow, nitrogen atoms in blue. (**a**) Semi-localized HOMO around N adjacent to a five-fold coordinated Si (**top left**). After trapping a hole, the state localizes at two of these N due to structural relaxations (**top right**), shifting the now unoccupied state near the middle of the band gap. (**b**) HOMO localized at twofold coordinated N (**top left**), with an additional hole localized at this site (**top right**). After trapping the hole, the amorphous network undergoes small structural relaxations and the state, now unoccupied, is shifted towards the middle of the band gap.

**Figure 4 nanomaterials-13-02286-f004:**
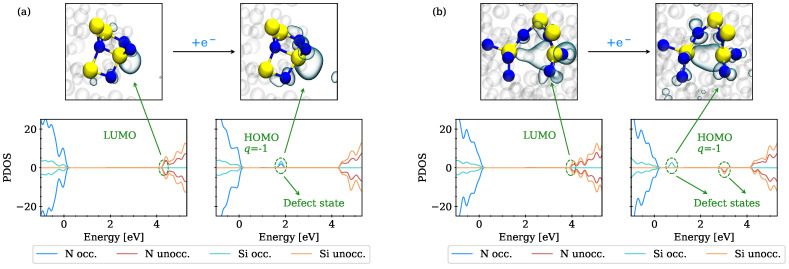
Electron trapping near intrinsic sites in a-Si${}_{3}$N${}_{4}$ with the localized wave functions (**top**) and the according projected density of states (**bottom**). Silicon atoms are shown in yellow, nitrogen in blue. (**a**) LUMO localized at Si adjacent to a four-fold coordinated N. After trapping an electron, the Si relaxes away from the N, thereby shifting the state, now occupied, towards the middle of the band gap. (**b**) LUMO hybridized between an undercoordinated and a fully coordinated Si (**top left**). An additional electron localizes at this site, thereby moving the undercoordinated Si towards the other (**top right**) and introducing one unoccupied and one occupied state in the band gap.

**Figure 5 nanomaterials-13-02286-f005:**
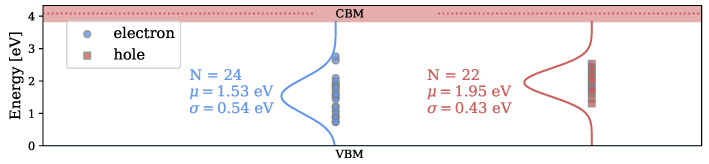
Energy levels of the electronic defect states introduced in the band gap after an electron (occupied state) or a hole (unoccupied state) is trapped in a-Si${}_{3}$N${}_{4}$.

**Figure 6 nanomaterials-13-02286-f006:**
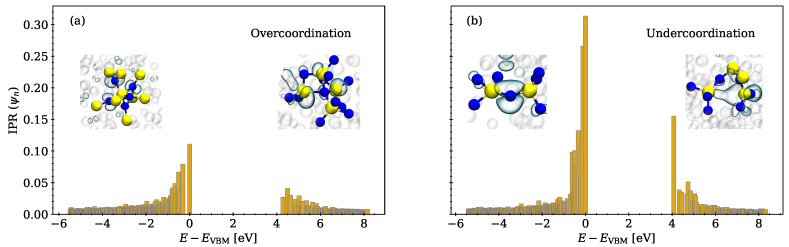
Inverse participation ratio of two different a-Si${}_{3}$N${}_{4}$ structures with the HOMO and the LUMO shown at an isovalue of 0.05 e/Å${}^{3}$ for the respective structures as an inset. Silicon atoms are shown in yellow, nitrogen atoms in blue. The IPR is a measure of the localization of an orbital, showing that the band edge states are (semi)localized. (**a**) States at the band edges introduced by overcoordinated atoms. (**b**) States at the band edges introduced by undercoordinated atoms or dangling bonds.

**Figure 7 nanomaterials-13-02286-f007:**
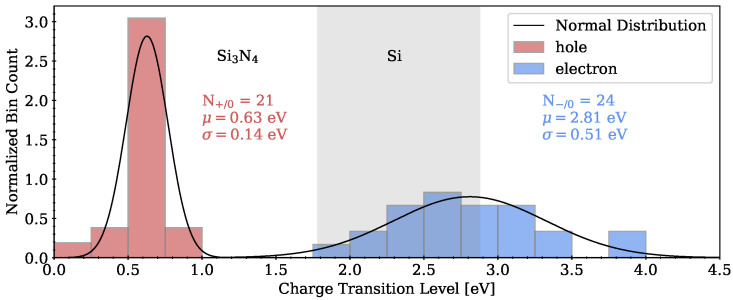
Charge transition levels of intrinsic hole and electron traps of several a-Si${}_{3}$N${}_{4}$ structures. CTLs are given with respect to the VBM of the according a-Si${}_{3}$N${}_{4}$ structure and shown in the context of a Si/Si${}_{3}$N${}_{4}$ band diagram with valence band offsets from experimental data.

**Figure 8 nanomaterials-13-02286-f008:**
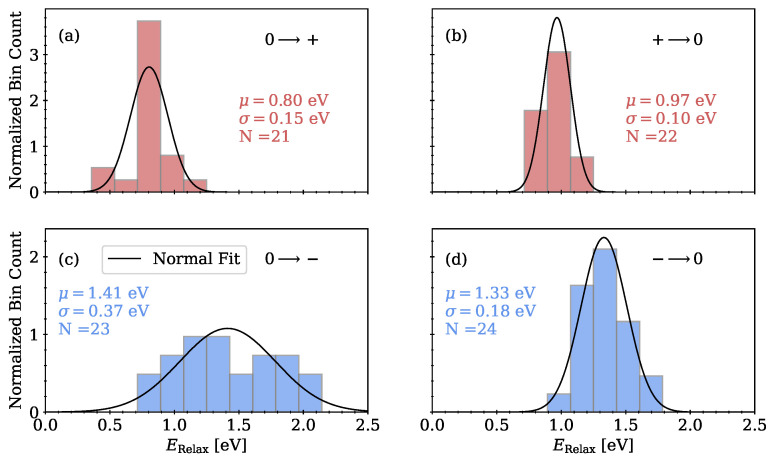
Relaxation energies according to the NMP model for different charge transfer processes with the fitting parameters of a normal distribution given in the plots. (**a**) Hole capture, (**b**) hole emission, (**c**) electron capture, (**d**) electron emission.

**Figure 9 nanomaterials-13-02286-f009:**
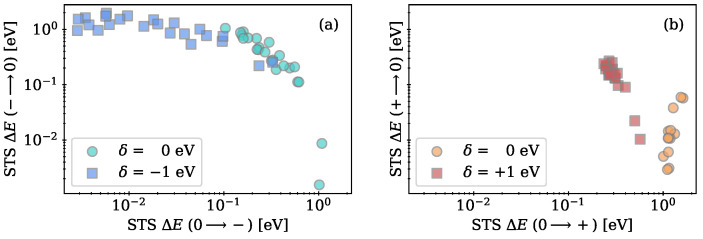
(**a**) Energy barriers in logarithmic scale from minimum energy configurations to the crossing point of the PECs for electron emission vs. electron capture with the CBM of a Si substrate acting as an electron reservoir. Energy values are shown for initial conditions and after shifting the trap level by δ=−1 eV after applying a positive voltage. (**b**) Energy barriers in logarithmic scale from minimum energy configurations to the crossing point of the PECs for hole emission vs. hole capture with the VBM of a Si substrate acting as a hole reservoir. Energy values are plotted for initial conditions and after decreasing the Fermi level by δ=+1 eV by applying a negative voltage.

## Data Availability

The data presented in this study are available on request from the corresponding author.

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
