# Peer review of "Over- and Undercoordinated Atoms as a Source of Electron and Hole Traps in Amorphous Silicon Nitride (a-Si3N4)"

_nanomaterials, 2023, doi:10.3390/nano13162286_

Round 1
Reviewer 1 Report
The authors approach the source of electron and hole traps in amorphous silicon nitride (a-Si3N4) by employing classical molecular dynamics, and a machine-learned interatomic potential, as well as DFT. Modeling details and choices are highly adequate and well-chosen, especially having in mind the amorphous materials system studied, and the bonding/electronic properties examined. Thus, the results look credible besides being very well presented. Equally important, the discussion is feasible and fits the current research questions regarding the electronic properties of amorphous materials.
All in all, this work represents a valuable contribution with possible wider impact in the field of thermal properties of nanostructures made out of wide bandgap semiconductors.
The authors chose an adequate structure of the manuscript. Concise, and nicely illustrated figures and their corresponding analysis are provided.
There are some minor issues with this already excellent manuscript that will need to be addressed before the manuscript becoming suitable for publication, i.e., it can be considered for publication after a minor revision:
1: Title: It is incorrect in English to begin every word in a title with a capital letters.
2: Abstract should briefly mention the methods and levels of theory used (especially in the case of DFT).
3: Simulation temperatures should be more clearly and explicitly mentioned throughout the text.
4: In the introduction, the authors should also mention that also ab molecular dynamics have been applied to structural and electronic property aspects of nanostructures of similar complexity, namely [Nanotechnology 33 (2022) 335706; Physical Chemistry Chemical Physics 25 (2023) 829-837]. Such works should be acknowledged thus providing a background of the present work.
5: Spell-check and stylistic revision of the paper are necessary. Some long sentences, as well as misspellings, etc., are noticeable throughout the text.
Spell-check and stylistic revision of the paper are necessary. Some long sentences, as well as misspellings, etc., are noticeable throughout the text.
Reviewer 2 Report
The analyzed work titled "Over- and Undercoordinated Atoms as a Source of Electron and Hole Traps in Amorphous Silicon Nitride (a-Si3N4)" investigates the charge trapping sites in a-Si3N4 and their impact on the memory effect in charge trap flash (CTF) devices. The study utilizes a combination of computational techniques, including machine-learned force fields and density functional theory calculations, to analyze the structural properties and electronic states of a-Si3N4.
The findings of this study, partcularly, provide insights into the behavior of charge trapping in a-Si3N4 and its relevance to CTF devices. The obtained parameters, including energy barriers and charge trapping characteristics, are valuable for developing device-level models and optimizing the design of future non-volatile flash memory devices.
There may be a possibility of publishing the work after addressing the following technical considerations:
Major
A. When using a machine-learned force field trained on data from DFT to create a-Si3N4 structures, there can be several issues related to accurately sampling the potential energy surface which are not properly discussed or noted in the present work. Some of these issues include:
-
Transferability of the Force Field: Machine-learned force fields are trained on a specific dataset and are designed to capture the behavior of atoms and molecules within the range of training data. However, the accuracy and transferability of the force field to predict the properties of a-Si3N4 may vary. The force field might not fully capture the complexity of the material or the intricacies of the chemical bonding, leading to limitations in sampling the potential energy surface accurately.
-
Representation of Structural Variations: Amorphous materials like a-Si3N4 can exhibit a wide range of structural variations, including different coordination environments and local atomic arrangements. Machine-learned force fields may struggle to adequately represent and sample these structural variations, leading to potential biases in the generated structures.
-
Missing Physical Effects: Machine-learned force fields often approximate the potential energy surface based on the training data. They may not fully account for all the relevant physical effects, such as charge transfer, polarization, or long-range interactions. This limitation can impact the accuracy of sampling the potential energy surface and may result in deviations from the true energetics of a-Si3N4.
B. Although the validity of the results is not criticized, I'm curious that when analyzing the charge trapping sites such as in a-Si3N4 using the nonradiative multi-phonon (NMP) model, there can be certain issues that need to be considered:
-
Model Assumptions: The NMP model makes certain assumptions about the nature of charge trapping and emission processes in amorphous materials. These assumptions may not fully capture all the complexities and nuances of charge trapping in a-Si3N4. It is important to recognize that the model provides an approximation and simplification of the underlying physical processes involved.
-
Accuracy of Parameters: The accuracy of the NMP model heavily depends on the accuracy of the input parameters, such as relaxation energies and charge transition levels. Estimating these parameters can be challenging, and inaccuracies in their determination may impact the reliability of the NMP model's predictions. It is crucial to ensure that the parameters are obtained from reliable and accurate sources, such as experimental measurements or high-level theoretical calculations.
-
Phonon Modes and Energy Transfer: The NMP model relies on the assumption that charge trapping and emission involve phonon-mediated processes. However, accurately describing the phonon modes and their interaction with charge carriers in a-Si3N4 can be complex. The effectiveness of energy transfer between charge carriers and phonons may vary, affecting the accuracy of the NMP model's predictions.
-
Role of Defect Structures: The NMP model assumes a simplified description of the defect structures and their electronic properties in a-Si3N4. However, the actual charge trapping sites in a-Si3N4 can be diverse and complex, with variations in local atomic arrangements and defect configurations. The model's ability to accurately capture the behavior of specific defect structures and their influence on charge trapping needs to be carefully evaluated.
C. The scientific and technical discussion of the results is insufficient in the current work. Instead of engaging in in-depth analysis and interpretation, the authors mainly state or reiterate their observations. Moreover, the Discussions and Conclusions section lacks appropriateness and fails to adequately describe the findings of the study. It is highly recommended that the authors enhance the manuscript by providing comprehensive and substantive physical discussions for each of the results obtained.
Minor
1. Experimental Validation: The study primarily relies on computational techniques, such as simulations and density functional theory calculations. It would be beneficial to validate the findings through experimental measurements (beyond Fig. 1) or additional characterization techniques.
2. Comparison with Existing Literature: It would be valuable to compare the results and findings of this study with existing literature on amorphous silicon nitride, intrinsic charge trapping sites, and their impact on CTF devices. This will help contextualize the novelty and significance of the research.
3. Addressing Limitations: The study should admit and discuss any limitations or assumptions made during the research. This could include limitations related to the chosen computational methods or the simplifications made in the model structures.
4. Again, separate Conclusions from the Discussions
5. Expand details about the theoretical approach MD, DFT, etc
6. Figure 8c seems to be underfitted
7. It is not clear the bin widths for each hystrogram
8. The pFIT3 auxiliary basis set may not fully capture the intricacies of the charge distribution, bonding, and defect states within a-Si3N4. The approximation could potentially lead to inaccuracies in the calculated electronic energies, transition levels, and charge trapping properties. The reliability and significance of the results obtained using pFIT3 for a-Si3N4 should be assessed by benchmarking against more accurate methods. Additionally, the sensitivity of pFIT3 to the system size and geometry of a-Si3N4 is another critical consideration. Amorphous materials like a-Si3N4 can have structural variations and disorder, which may affect the performance of the pFIT3 auxiliary basis set. Size consistency issues and the treatment of basis set superposition errors should be carefully evaluated, as they can impact the accuracy of the calculated energies and the analysis of charge trapping phenomena.
Reviewer 3 Report
Wilhelmer et al. investigated the origin of electron and hole traps in amorphous silicon nitride using a combination of the density-functional theory and machine-learning force field approach. Both the first-principles code CP2K and the classic molecular dynamics package LAMMPS have been well-established. The design of the systems and the related computational settings are reasonable. The obtained results are useful for people working in semiconductor physics. The manuscript is written properly. The text is in the scope of this Journal. It is my pleasure to suggest acceptance of this manuscript for publication in Nanomaterials in the present form.
Round 2
Reviewer 2 Report
It is appreciated considering comment and suggestions by Authors. However, from my side, the Conclusions must be separated from Discussion section.
In my opinion, the absent of discussions that shed light on the results is noticeable.
Author Response
We thank the reviewer for their suggestions to further improve our manuscript.
Nevertheless, we would like to point out, that thorough discussions of the results are already given throughout the "Results" section. This includes comparison with experimental data and theoretical results from literature and discussion of the localization of defect states and their impact on non-volatile memory devices within the nonradiative multi-phonon model.
We suggest changing the title of Section 3 from "Results" to "Results and Discussion" to emphasize the scope of the following section. Hereby, we follow the guidelines of the MDPI template, that states that an extra section for Discussion and Conclusion is only optional. This structure of the manuscript is also in accordance with recent publications in "Nanomaterials" that discuss the results during their presentation and give the conclusions at the end to summarize the paper.